# Dose Optimization of Colistin: A Systematic Review

**DOI:** 10.3390/antibiotics10121454

**Published:** 2021-11-26

**Authors:** Abdul Haseeb, Hani Saleh Faidah, Saleh Alghamdi, Amal F. Alotaibi, Mahmoud Essam Elrggal, Ahmad Jamal Mahrous, Safa S. Almarzoky Abuhussain, Najla A. Obaid, Manal Algethamy, Abdullmoin AlQarni, Asim A. Khogeer, Zikria Saleem, Aziz Sheikh

**Affiliations:** 1Department of Clinical Pharmacy, College of Pharmacy, Umm Al Qura University, Makkah 24382, Saudi Arabia; afotaibi@uqu.edu.sa (A.F.A.); merggal@uqu.edu.sa (M.E.E.); ajmahrous@uqu.edu.sa (A.J.M.); ssmarzoky@uqu.edu.sa (S.S.A.A.); 2Department of Microbiology, Faculty of Medicine, Umm Al Qura University, Makkah 24382, Saudi Arabia; hsfaidah@uqu.edu.sa; 3Department of Clinical Pharmacy, Faculty of Clinical Pharmacy, Al Baha University, Al Baha 65779, Saudi Arabia; saleh.alghamdi@bu.edu.sa; 4Department of Pharmaceutics, College of Pharmacy, Umm Al Qura University, Makkah 24382, Saudi Arabia; naobaid@uqu.edu.sa; 5Alnoor Specialist Hospital, Department of Infection Prevention & Control Program, Makkah 24382, Saudi Arabia; mmalgethamy@moh.gov.sa; 6Alnoor Specialist Hospital, Infectious Diseases Department, Makkah 24382, Saudi Arabia; al-qrni@hotmail.com; 7Plan and Research Department, General Directorate of Health Affairs of Makkah Regiona, Ministry of Health, Makkah 24382, Saudi Arabia; akhogeer@moh.gov.sa; 8Medical Genetics Unit, Maternity & Children Hospital, Makkah Healthcare Cluster, Ministry of Health, Makkah 24382, Saudi Arabia; 9Department of Pharmacy Practice, Faculty of Pharmacy, The University of Lahore, Lahore 54000, Pakistan; xikria@gmail.com; 10Usher Institute, The University of Edinburgh, Teviot Place, Edinburgh EH16 4UX, UK; aziz.sheikh@ed.ec.uk

**Keywords:** dose optimization, colistin, Gram-negative infections, nephrotoxicity

## Abstract

Colistin is considered a last treatment option for multi-drug and extensively resistant Gram-negative infections. We aimed to assess the available data on the dosing strategy of colistin. A systematic review was performed to identify all published studies on the dose optimization of colistin. Grey literature and electronic databases were searched. Data were collected in a specified form and the quality of the included articles was then assessed using the Newcastle-Ottawa scale for cohort studies, the Cochrane bias tool for randomized clinical trials (RCT), and the Joanna Briggs Institute (JBI) critical checklist for case reports. A total of 19 studies were included, of which 16 were cohort studies, one was a RCT, and two were case reports. A total of 18 studies proposed a dosing regimen for adults, while only one study proposed a dosing schedule for pediatric populations. As per the available evidence, a loading dose of 9 million international units (MIU) of colistin followed by a maintenance dose of 4.5 MIU every 12 h was considered the most appropriate dosing strategy to optimize the safety and efficacy of treatment and improve clinical outcomes. This review supports the administration of a loading dose followed by a maintenance dose of colistin in severe and life-threatening multi-drug Gram-negative bacterial infections.

## 1. Introduction

Antibiotic resistance is a global public health and clinical challenge [1]. Many expert reports and clinical guidelines have been published to highlight and address this concern [2]. Despite this awareness, antibiotic resistance remains a persistent problem all over the world [3]. The increasing trend of multi-drug resistance in Gram-negative bacteria (MDR-GNB) poses a particularly acute challenge to health systems.

Polymyxin antibiotics (Polymyxin E and Polymyxin B) are polypeptide antibiotics that show significant activity against the Gram-negative bacterial infections caused by *Pseudomonas aeruginosa*, *Acinetobacter baumannii*, *Klebsiella pneumoniae*, and carbapenem-resistant Enterobacteriaceae [4,5]. Colistin (also known as Polymyxin E) was the first polymyxin antibiotic available on the market in 1950s [6,7]. It is administered as colistimethate sodium (CMS), an inactive drug that converts to an active moiety of colistin base activity (CBA) on hydrolysis [8,9]. However, due to the nephrotoxicity caused by colistin, it was replaced by other alternative therapies with fewer side effects [10]. In the past few years, the increasing trend of MDR-GNB and lack of alternative treatment options has led to the reintroduction of colistin for clinical use [11]. Since the reintroduction of colistin in clinical practice, a limited number of articles have been published regarding its pharmacokinetic and pharmacodynamic (PK/PD) parameters, especially in critically ill patients, to optimize the plasma concentration for killing bacteria and minimize the risks of nephrotoxicity [12].

Colistin has an extremely narrow therapeutic index (2–4 mg/L). The desired plasma concentration required for an anti-bacterial effect may overlap with the concentration that predisposes it to nephrotoxicity [13]. As suggested by international consensus guidelines for the optimal use of colistin, dose optimization at an individual patient level is essential [14]. This strategy requires that real-time data on pharmacokinetic profiles are obtained from patients during colistin therapy [15]. Being an old drug, colistin was never subjected to the regulatory approval procedures required for modern drug development. [16]. However, no scientifically-based dosage strategy is available for patients, particularly critically ill patients who are receiving renal replacement therapy and patients with cystic fibrosis [17].

No new antibiotics are scheduled to be introduced for the treatment of Gram-negative bacterial infections within the next few years. There is therefore an urgent need to optimize the use of the available antibiotics [18]. Several approaches for a colistin dosing regimen have been adopted, including a loading dose followed by a maintenance dose, a higher dose as per patient renal function and targeted steady state concentration of colistin, local administration (intraventricular or inhaled), and antibiotic combination therapy [19].

As per colistin’s pharmacokinetics/pharmacodynamics (PK/PD) profile and the observed high interpatient variability in plasma concentrations, specific dosage forms, e.g., intravenous, inhalational, and intraventricular, are recommended in critically ill patients depending on the targeted site of infection [20]. A better understanding of all of these parameters is needed for the optimization of the clinical use of colistin. Therefore, this systematic review aimed at mapping the available literature on the interventions used to address the dosing strategy of colistin.

## 2. Results

### 2.1. Study Selection

A total of 453 related published articles were retrieved from grey literature and electronic databases. After the deletion of duplicates, 336 articles were assessed for eligibility. Based on the exclusion and inclusion criteria, 240 articles were excluded after the screening of the titles and abstracts of selected studies. Several studies were retrieved from reference lists of included studies and other systematic reviews. After screening of full-text articles, 78 articles were excluded for the following reasons: literature reviews (*N* = 5), redundant publications (*N* = 23), inappropriate intervention (*N* = 8), non-English (*N* = 21), and not having the required data (*N* = 20).

Nineteen articles met the inclusion criteria for this systematic review. All included studies were published in English. The main characteristics of the selected studies are listed in Table 1 [12,17,21,22,23,24,25,26,27,28,29,30,31,32,33,34,35,36]. Only one of these studies was conducted in a pediatric population. The eligible studies were published from 2005 to 2016. Of the 19 articles, 16 were cohort studies, one was a RCT, and two were case reports. A total of 860 patients were recruited in the articles (Figure 1).

### 2.2. Quality Assessment of Studies

Our assessment of study quality is summarized in Table 2, Table 3 and Table 4. Based on the NOS, nine studies were rated as a total score of seven, one study scored six, while the remaining six studies scored eight. Overall, the score of the included studies was seven. The Cochrane bias tool evaluated that most of the domains for RCT were at low risk of bias. According to the JBI critical checklist for assessing the quality of case reports, both studies were of good quality.

### 2.3. Dose Optimization of Colistin

In this systematic review, 19 articles involving 860 patients with Gram-negative infections provided data on the dosing strategy of colistin. A dose of CBA of 2.5–5 mg/kg/day was usually recommended by manufacturers in the USA [37]. Most included studies used the recommended loading dose of colistin, followed by the maintenance dose, a regimen that was found not to result in significant renal and nephrological toxicity [21,22,23,25,26,27,28].

Karaiskos et al. studied 19 critically ill patients, who received a loading dose of 9 million international units (MIU) of colistin, followed by a maintenance dose of 4.5 MIU every 12 h, and overall the incidence of acute renal injury was 20% [21]. They reported that the dose should be adjusted according to renal function. A similar study published by Dalfino et al. included 28 patients with Gram-negative infections [23]. The loading dose of 9 MIU of colistin followed by a maintenance dose of 4.5 MIU every 12, 24, or 48 h also resulted in clinical cure in 82%. Elefritz et al. included 72 patients, who (*N* = 30) received a dose of colistin according to renal function in a loading dose group versus standard dose group (*N* = 42) without loading dose [26]. Positive clinical outcomes were achieved in 55% of patients in the standard group, while 67% of the patients showed clinical cure in the loading dose group. Similarly, Trifi and his colleagues also compared the dosing regimen with and without a loading dose [25]. He reported that 63% of the patients were cured in the group with a loading dose.

Dalfino and her colleagues also published another article documenting that concomitant administration of ascorbic acid minimizes the risk of acute kidney injury (AKI), permitting the safer use of colistin [24]. Seven studies included 329 patients, in which the dosing strategy did not include a loading dose [12,31,32,33,34,35,36]. DeRyke and his colleagues reported that colistin-associated nephrotoxicity developed in 33% of the patients [35]. He suggested the dose of colistin should be adjusted by using a measure of lean body mass, such as ideal body weight (IBW), to lower the risk of colistin-related nephrotoxicity. Jung et al. included 153 patients, who received an inhalational dose of colistin prior to IV colistin therapy, which resulted in reducing the risk of colistin associated nephrotoxicity [29]. Javan and his colleagues conducted RCT, reporting the highest prevalence rate of nephrotoxicity in the high dose group (60%) compared to the conventional dose group (20%). Similarly, significant variability in colistin concentration was observed in both case reports, resulting in the high risk of colistin-associated nephrotoxicity [34,36].

**Table 1 antibiotics-10-01454-t001:** Dose optimization of colistin.

Author and Year	Study Design	Sample Size	Characteristics of Patients	Dosing Practice	Clinical Outcomes	Dosing Recommendation
Karaiskos, 2015 [21]	Multi-center prospective study	19	Patients with VAP, tracheobronchitis, bacteremia, intra-abdominal acute pyelonephritis infections	LD of 9 MIU followed by MD of 4.5 every 12 h.	20% of patients developed acute renal injury.	Patients with Clcr >80 mL/min/1.73 m^2^ required high dose of MD to achieve colistin concentration above 2 mg/L at steady state.
Garonzik, 2011 [17]	Prospective study	105	Patients with BSI and pneumonia	The median daily dose of colistin base was 200 mg.	The recommended dose did not achieve adequate colistin/CMS plasma concentration.	Colistin/CMS may be used as a combination therapy for positive clinical outcomes.
Javan, 2017 [38]	RCT	40	Patients with MDR-GNB infections	High dose group: LD of 9 MIU followed by MD of 4.5 every 12 h. Conventional dose group: A Dose of 2 MIU every 8 h.	The prevalence of nephrotoxicity was higher in the high dose group (60%) as compared to conventional group (20%).	More RCT are recommended on a large scale to identify the optimal dosing strategy.
Gregoire, 2014 [22]	Multi-center population kinetic study	73	Patients with Gram-negative infections	The median LD was 2 MIU followed by median MD of 6 MIU/day CMS was also aerosolize 1–2 MIU 1 or 3 times daily.		MD should be adjusted according to renal function.
Dalfino, 2012 [23]	Prospective cohort study	28	Patients with Gram-negative infections	LD of 9 MIU followed by MD of 4.5 every 12 h for Clcr < 50 mL/min/1.73 m^2^. MD of 4.5 MIU after 24 h for Clcr 20–50 mL/min/1.73 m^2^ MD of 4.5 MIU after 48 h for Clcr < 20 mL/min/1.73 m^2^.	The incidence rate of clinical cure was 82%.	A 9 MIU twice daily dosing regimen of colistin, along with a 9 MIU loading dose can result in positive clinical outcomes, with no or fewer side effects.
Dalfino, 2015 [24]	Prospective cohort study	70	Patients with VAP, BSI, UTIs and sepsis	For Clcr 60–130 mL/min/1.73 m^2^, a daily dose of fixed dose of 9 MIU was administered. For Clcr > 130 mL/min/1.73 m^2^, a daily dose of 10–12 MIU was allowed. MD was adjusted every 12 h after LD.	56% showed positive clinical outcomes while 44% developed AKI.	Concomitant administration of ascorbic acid minimizes the risk of AKI, thus permitting safer and effective use of colistin.
Trifi, 2016 [25]	Prospective comparative study	92	Patients with VAP, CRI	1st group: LD of 9 MIU followed by MD of 4.5 every 12 h. 2nd group: A dose of 6 MIU colistin was administered.	63% of the patients were cured in the higher dose group.	The high dose of colistin regimen is more effective, with relatively low colistin associated nephrotoxicity.
Elefritz, 2017 [26]	Retrospective cohort study	72	Patients with pneumonia	Pre-implementation group: GFR > 70 mL/min, a dose of 2.5 mg/kg every 12 h GFR 30–70 mL/min, a dose of 1.5 mg/kg every 24 h GFR < 30 mL/min, a dose of 1.5 mg/kg every 48 h Post-implementation group Loading dose of 5 mg/kg GFR > 50 mL/min, a dose of 3.5 mg/kg every 12 h GFR 20–50 mL/min, a dose of 3.5 mg/kg every 24 h GFR < 20 mL/min, a dose of 3.5 mg/kg every 48 h.	The incidence rate of Clinical cure was 55% for patients in the pre-implementation group, while 67% patients in post-implementation group.	LDHD dosing regimen is associated with significant clinical or microbiological benefits.
Hengzhuang, 2017 [27]	Prospective study	10	Patients with pulmonary infections	Doses of CMS of 6 MIU and 9 MIU were administered by intravenous infusion over 45 and 90 min.	The PTA was 49.8%, 53.8%, and 99.4% for planktonic infection, and 11.3%, 14.6%, and 65.3%, respectively, for biofilm infection.	Colistin dosage of 9 MIU is better than 6 MIU for planktonic as well as biofilm infections of *P. aeruginosa*
Wacharachaisurapol, 2020 [28]	Prospective, open label	20	Patients with Gram-negative infections	loading dose (LD group) of 4 mg of colistin base activity (CBA)/kg/dose or a standard initial dose (NLD group) of 2.5 mg (12 h interval) or 1.7 mg (8 h interval) of CBA/kg/dose.	no patient in either group experienced AKI.	A higher daily dose of CMS should be considered for the treatment of MDR-GNB infections.
Jung, 2019 [29]	Retrospective	153	Patients with pneumonia and bacteremia	The average daily dose of IV colistin is 312 mg. Patients also received inhaled colistin therapy.	Colistin-associated nephrotoxicity was substantially less likely to develop in patients who received inhaled colistin close to the time of IV colistin therapy.	Use of inhaled colistin immediately prior to the initiation or after the end of systemic colistin therapy maximizes the therapeutic effectiveness.
Marin, 2016 [30]	Prospective	100	Patients with VAP	2 MIU of CMS three times daily.	This dosing recommendation reported efficacy in 94.6% patients with VAP.	MDR AB treated with colistin does not have lower mortality rates than previous studies.
Imberti, 2010 [31]	Prospective, open label study	13	Patients with VAP	A dose of 2 MIU of CMS (174 mg) q8h given IV for at least 2 days.	The recommended dose of CMS resulted in suboptimal plasma concentration of colistin with no nephrotoxicity.	IV administration of recommended dose of CMS is effective for the treatment of MDR Gram-negative infections.
Markou, 2008 [32]	Prospective, open label study	14	Patients with sepsis	IV administration of 225 mg CMS every 8 h or 12 h after infusion.	Colistin related nephrotoxicity was not observed.	CMS dosage regimen administered were associated with suboptimal Cmax/MIC ratios for many Gram-negative pathogens currently reported as sensitive.
Plachouras, 2009 [33]	Prospective	18	Patients with Gram-negative bacterial infections	IV administration of CMS of dose of 3 MIU (240 mg) every 8 h.	Plasma colistin concentration was insufficient before steady state.	Change in the dosing strategy for colistin may be needed.
Li, 2005 [34]	Case report	1	Patient receiving continuous venovenous hemodiafiltration	IV administration of CMS of 150 mg every 24 h on day of 24, IV administration of CMS of dose 150 mg every 48 h on day of 38.	Plasma concentration of colistin and CMS was below the respective MICs approximately 4 h following administration of CMS.	The dosage of CMS should be modest, i.e., 2–3mg/mg every 12 h.
DeRyke, 2009 [35]	Retrospective, cohort study	30	Patients with Gram-negative bacterial infections	IV administration of colistin of 5.1 ± 2.4 mg/kg/day.	33% of patients developed nephrotoxicity.	Using a measure of lean body mass such as IBW to dose colistin may be less nephrotoxic.
Akers, 2015 [36]	Case report	02	Burn patient receiving venovenous hemodiafiltration	Patient 1: IV CMS (2.2 mg CBA/kg every 12 h, infused over 30 min) and nebulized CMS (75 mg every 8 h). Patient 2: CMS was infused over 30 min at 2.9 mg CBA/kg/day (in 2 divided doses) initially and increased to 4.4 mg CBA/kg/day (2 divided doses) after CVVH was prescribed at 35 mL/kg/h. Inhaled CMS was at 75 mg every 8 h.	We observed significant variability in colistin concentrations, resulting from recommended dosing strategies reporting the risk of for toxicity and compromised PK/PD target attainment.	PK/PD data of colistin is required, particularly for those undergoing continuous renal replacement therapy.
Ram, 2021 [12]	Prospective open label study	30	Patients with Gram-negative infections	IV CMS of dose of 2 MIU with inhalational CMS 1 MIU every 8 h.	Of 30 patients, 20 patients showed clinical improvement.	Future large scale studies are warranted, to shed further light on the role of various PK/PD parameters of colistin, in order to devise or select an optimal dosing strategy.

LD: loading dose, MD: maintenance dose, BSI: blood stream infections, UTI: urinary tract infections, VAP: ventilator-acquired pneumonia, CRI: catheter related infections, CLcr: creatinine clearance, Cmax: maximum plasma concentration, AKI: acute kidney injury, CMS: colistimethate sodium, CBA: colistin base activity, PTA: patient target attainment, MIU: million international unit, MDR-GNB: multi-drug resistant Gram-negative bacteria; IV: intravenous; IBW: ideal body weight, LDHD: loading dose higher dose.

**Table 2 antibiotics-10-01454-t002:** Quality assessment of cohort studies.

	Selection				Comparability	Outcomes		
Reference	Representative of Exposed Studies ^A^	Selection of Non-Exposed ^B^	Ascertainment of Exposure ^C^	Demonstration of Outcome ^D^	Comparability of Cohort Studies on Basis of Design ^E^	Assessment of Outcomes ^F^	Adequacy of Follow-Up ^G^	Quality Score
Karaiskos, 2015 [21]	*	*	*	*	*	*	*	7
Garonzik, 2011 [17]	*	*	*	*	*	*	*	7
Gregoire, 2014 [22]	*	*	*	*	*	**	*	8
Dalfino, 2012 [23]	*	*	*	*	*	**	*	8
Dalfin, 2015 [24]	*	*	*	*	*	**	*	8
Trifi, 2016 [25]	*	*	*	*	*	**	*	8
Elefritz, 2017 [26]	*	*	*	*	*	**	*	8
Hengzhuang, 2017 [27]	*	*	*	*	*	**	*	8
Wacharachaisurapol, 2020 [28]	*	*	*	*	*	*	*	7
Jung, 2019 [29]	*	*	*	*	*	*	*	7
Marin, 2016 [30]	*	*	*	*	*	*	*	7
Imberti, 2010 [31]	*	*	*	*	*	**	*	8
Markou, 2008 [32]	*	*	*	*	*	*	*	7
Plachouras, 2009 [33]	*	*	*	*	*	*	*	7
DeRyke, 2009 [35]	*	*	*	*	*	*	*	7
Ram, 2021 [12]	*	*	*	*	*	*	-	6

A: * = truly representative or somewhat representative of average in target population. B: * = drawn from the same community. C: * = secured record or structured review. D: * = yes, - = no. E: * = study controls for age, gender, and other factors. F: * = record linkage or blind assessment, ** = Both. G: * = follow-up of all subjects.

**Table 3 antibiotics-10-01454-t003:** Risk of bias assessment for randomized controlled trials.

Study	Random Sequence Generation	Allocation Concealment	Blinding of Participants and Personnel	Blinding of Outcome Assessment	Incomplete Outcome Data	Selective Reporting	Other Bias
Javan et al., 2017 [38]	Low risk	Low risk	Unclear	Unclear	Low risk	Low risk	Unclear

**Table 4 antibiotics-10-01454-t004:** Quality assessment of case reports.

Study	Q1	Q2	Q3	Q4	Q5	Q6	Q7	Q8	Quality Rating
Li, 2005 [34]	Yes	Yes	Yes	Yes	Yes	Yes	Yes	Yes	Good
Akers, 2015 [36]	Yes	Yes	Yes	Yes	Yes	Yes	No	Yes	Good

## 3. Discussion

Rational antibiotic dosing is essential for maximizing therapeutic effectiveness, with no or fewer side effects [39]. The manufacturer suggests a traditional treatment regimen, which is adopted by physicians and veterinary surgeons while prescribing the antibiotics [40]. In these treatment regimens, the fixed-dose of antibiotics was administered for the specified time-period. Several drug efficiency studies were reported in a systematic review to evaluate the dose and duration of these treatment regimens [41]. However, these only provide data for the regimen being evaluated, and no data are available for the indication of other possible regimens. A study documented the significance of the appropriate use of antibiotics and the need to incorporate PK/PD data into dosage scheduling [42]. This review provided the available evidence evaluating the dosing strategy of colistin among critically ill patients. Regarding the changes in dosing strategies, studies showed that a loading dose may help to minimize the risk of colistin-associated nephrotoxicity. Front loading is considered the optimal dosing strategy, which maximized the therapeutic outcomes and minimized the risk of side effects [43]. The European Society for Clinical Microbiology and Infectious diseases (ESCMID) recommended a daily dose of colistin base activity (CBA) of 9–10.9 MIU, divided into two and infused over 0.5–1 h at a 12-h interval in patients with normal renal function, in order to achieve the desired therapeutic outcomes. The CBA dose adjustment is made according to creatinine clearance in patients with renal impairment [15].

Inadequate dosing should be considered as a primary reason when positive clinical outcomes were not achieved in patients with suspected or documented Gram-negative infections [44]. Therefore, dose optimization of colistin should be considered in both the initial dosing and dose adjustment. A study reported that colistin can take more than 36 h to reach a steady-state plasma concentration of 2 mg/L upon administration of 3 MIU CMS every 8 h in patients with normal renal function [33]. This result highlights that low initial exposure to colistin formation is a substantial PK/PD challenge for optimizing the use of CMS in patients [45]. This issue can be partially resolved with the use of a loading dose. A study reported that after the administration of loading doses of 6 MIU and 9 MIU, the average colistin plasma concentration reached 1.34 mg/L and 2.65 mg/L, respectively, at 8 h after the loading dose; with increased likelihood of earlier eradication of the infecting bacteria [21,46]. Karaikos and his colleagues also reported that the administration of a loading dose of 9 MIU CMS followed by a maintenance dose in critically ill patients with Gram-negative infections achieved a steady-state colistin concentration above the breakpoints, which resulted in fast clinical improvement [21].

A study published by Dalfino also documented that a high dose with extended intervals CMS regimen is highly effective, without significant renal toxicity [23]. However, a study reported by Elefritz et al. documented that the clinical rates were not improved after implementation of loading dose high dose (LDHD) guidelines [26]. The AKI was reported to be higher (58%) in the post-implementation group. Similarly, Trifi and colleagues also reported that no higher risk of nephrotoxicity was found when increasing daily doses of colistin [25].

Renal dysfunction substantially alters the pharmacokinetic parameters of a drug [47]. An inappropriate dose adjustment could result in elevated plasma concentration of a drug, leading to adverse drug reactions, and thus increasing the risk of mortality, morbidity, and increased length of hospital stay [48]. Nephrotoxicity and neurotoxicity are the most substantial and frequent adverse effects of polymixins [49]. In pediatrics, the loading dose of 4mg of CBA/kg is beneficial for the improvement of drug exposure, by increasing the area under curve (AUC) and maximum plasma concentration (C_max_) without AKI [28]. The prevalence of colistin-associated nephrotoxicity ranges from 0.6% to 10% in the pediatric population [50]. On the basis of RIFLE (risk, injury, failure, loss of kidney function, end-stage kidney disease) criteria, a study reported a high prevalence rate of nephrotoxicity in a higher dose group (60%) than in a conventional dose group (20%) [28]. Similarly, another study documented that 44% of the patients developed AKI after implementation of a loading dose followed by a maintenance dose [24]. However, most of the studies reported less or no nephrotoxicity associated with higher doses [23,25,31,32]. According to this extensive systematic review, there was huge variation between articles, due to small sample sizes, time of assessment, colistin dosing regimens, how missing data were secured, and the criteria for outcome assessment. Due to limited PK/PD data, the optimal dose of colistin has not yet been evaluated in critically ill patients. For a better understanding of the data, more randomized controlled trials are required to redefine the rational dosing of colistin. This strategy concerns all potential routes of colistin administration, to improve the colistin clinical efficacy with fewer adverse effects [51].

This systematic review has some limitations that should be acknowledged. Primarily, limited databases were utilized, focusing on the titles describing the dose optimization of colistin, and no quantitative analysis was performed. Second, there was a small sample size in each included study, which could have altered the reliability and validity of the clinical outcomes. Despite these limitations, we consider that this systematic review explores the relevant articles on this topic, allowing us to focus on the dosing strategy of colistin in patients.

## 4. Methods

### 4.1. Data Sources and Searches

The systematic search assessing the interventions on dose optimization of colistin was conducted according to preferred items for systematic reviews and meta-analysis (PRISMA) guidelines [52]. The two reviewers independently searched published and grey literature. The main terms used in the search strategy were ‘colistin’ or ‘dose optimization’ or ‘pharmacokinetic’ or ‘pharmacodynamic’ or ‘drug administration’ or ‘dosage’ or ‘adults’ or ‘pediatrics’ or ‘nephrotoxicity’ or ‘therapeutic drug monitoring’.

### 4.2. Inclusion and Exclusion Criteria

Initially, the selection of articles obtained from the aforementioned terms was assessed by reading the title or abstract. All relevant articles were independently reviewed by two reviewers for the eligibility criteria. The articles retrieved from search items were merged, and duplicates were eliminated. Full-text articles on the dose optimization of colistin were included in this review. Although we did not impose any language barrier in the search items, only articles written in English were included in this study. In addition, conference abstracts were excluded, because they do not provide sufficient data to allow for evaluation. Moreover, the bibliographies of published reviews and the included studies were extensively reviewed to discover other potentially eligible studies. Additionally, to identify additional peer-reviewed articles, the reference lists of the selected articles were screened. The full-text was evaluated and included when they met the following pre-specified criteria: (1) dosing strategy of colistin; and (2) outcomes measures related to safety and efficacy of colistin. Differences were resolved through discussion or, if necessary, by discussing with a third reviewer. The type of studies included were cohort studies, case reports, and a randomized control trial.

### 4.3. Quality Assessment

The quality of the cohort studies was evaluated using Newcastle-Ottawa scale (NOS). The NOS is a nine-star rating system that classifies the data into three subscales, i.e., selection, comparability, and outcomes [53]. A maximum of 4 stars can be allotted in the item of selection, 2 stars in comparability, and 2 stars in outcomes. The Cochrane assessment tool was used to assess the randomized controlled studies (RCTs), by assessing the risk of bias in each study [54]. This tool is structured into domains that judged the RCTs based on ‘high risk’, ‘low risk’, and ‘unclear’. The Joanna Briggs institute (JBI) critical checklist was utilized to evaluate the quality of case reports [55]. Two reviewers independently assessed the methodological quality of each selected article. Reviewers compared their results and disagreements were then resolved by discussion. Information on quality assessment tools can be assessed in the Appendix A.

### 4.4. Data Extraction

Data were retrieved from text, tables, and figures from each included article and were noted in a pre-specified data collection form. This customized data form included study characteristics (author’s name, year of publication, design, and sample size) and patient characteristics (patient clinical condition, dosing regimen, outcomes of interests, and dosing recommendation). Data extraction forms can be assessed in the Appendix A. Data extraction was completed by one reviewer and was then reviewed by another reviewer. Disagreements were addressed by discussion between two reviewers or in consultation with the third reviewer, if necessary.

## 5. Conclusions

This review supports the administration of a loading dose followed by a maintenance dose of colistin in severe and life-threatening multi-drug Gram-negative bacterial infections. However, in obese patients, ideal body weight (IBW) is required to calculate the colistin dose. The current data indicate that intravenous along with aerosolized colistin do not potentiate nephrotoxicity. Due to limited data on clinical effectiveness, differences in outcomes may occur. Therefore, large-scale and well-designed studies are required to assess the dosing regimen of colistin, to provide insights into strategies to maximize the target therapeutic outcomes and optimize patient safety.

## Figures and Tables

**Figure 1 antibiotics-10-01454-f001:**
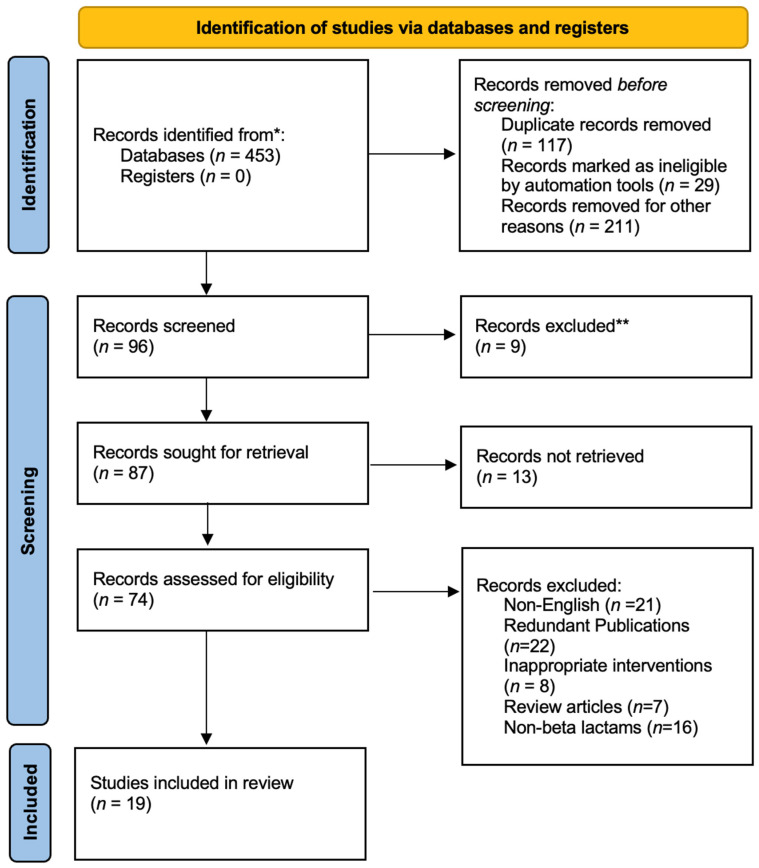
PRISMA flow diagram. * Consider, if feasible to do so, reporting the number of records identified from each database or register searched (rather than the total number across all databases/registers). ** If automation tools were used, indicate how many records were excluded by a human and how many were excluded by automation tools.

## Data Availability

Data is available within the article.

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
