# Peer review of "Dose Optimization of Colistin: A Systematic Review"

_antibiotics, 2021, doi:10.3390/antibiotics10121454_

Round 1

Reviewer 1 Report

The aim of this review is clearly described with appropriate scientific soundness. The introduction part is well written. Data presentation and description of the key findings were logically organized. Determination of optimized dose of colistin is a valuable aspect in terms of treatment of critically ill patients, the authors tried to contribute to the development of proper use of colistin.

1. The authors should have elaborated on the main objective of the research, specifically, previous literature suggests that colistin may not be highly toxic (Crit Care . 2006 Feb;10(1):R27), so a proper dosing strategy could reduce the adverse effect. the authors fail to address the bullet point clearly. 
2. The authors tried to set up the effective use of colistin based on available published data, but the source of data is weak.
3. The Sample sizes from data sources should have been bigger. The authors should have included other sources of data for credible output.
4. I think the overall conclusion is precise and brief

Author Response

The aim of this review is clearly described with appropriate scientific soundness. The introduction part is well written. Data presentation and description of the key findings were logically organized. Determination of optimized dose of colistin is a valuable aspect in terms of treatment of critically ill patients, the authors tried to contribute to the development of proper use of colistin.

Author’s Response:

Many thanks for providing positive feedback. We have corrected all comments by reviewer 1. Detail of each correction is given below.

Reviewer 1 Comment 1:

The authors should have elaborated on the main objective of the research, specifically, previous literature suggests that colistin may not be highly toxic (Crit Care . 2006 Feb;10(1):R27), so a proper dosing strategy could reduce the adverse effect. the authors fail to address the bullet point clearly. 

Authors’ response to comment 1: 

Thankyou for the valuable feedback. Colistin is an old antibiotic which was replaced by aminoglycosides because of reports of significant side effects such as nephrotoxicity and neurotoxicity. Due to increase in antimicrobial resistance, colistin is now considered as last-line treatment option for multi-resistant gram-negative bacterial infections. Therefore, this systematic review highlights the appropriate dosing strategy of colistin to maximize antibacterial efficacy and minimize the toxicity and development of the resistance. We hope this is clear.

Reviewer 1 Comment 2:

The authors tried to set up the effective use of colistin based on available published data, but the source of data is weak.

Authors’ response to comment 2: 

Thankyou for the feedback. We have used various databases like Google Scholar, PubMed, Scopus, and ScienceDirect as per PRISMA guidelines. All these databases are reliable research data sources. Secondly, the quality of the studies was also checked by using quality assessment tools such as New-Castle-Ottawa Scale for cohort studies, Cochrane assessment tool for RCT and Joanna Briggs institute (JBI) critical checklist for case reports. We hope this is acceptable.

Reviewer 1 Comment 3:

 The Sample sizes from data sources should have been bigger. The authors should have included other sources of data for credible output.

Authors’ response to comment 3: 

Thank you for the feedback. The comprehensive grey literature was also performed by using Google Scholar and other websites from healthcare services, CDC, WHO and IDSA. Secondly, peer-reviewed literature (Scopus, Science Direct, PubMed, and EMBASE) was also employed. The data on colistin is limited because of concerns of nephrotoxicity and neurotoxicity.

 Reviewer 1 comment 4:

 I think the overall conclusion is precise and brief.

Authors’ response comment 4

Thank you for the feedback. It has been revised. Some more explanations have been added.

Reviewer 2 Report

The authors systematically reviewed articles regarding the dosage of colistin, and stated the nephrotoxicity by colistin. This review is useful in the clinical settings due to a lack of data regarding the optimal dosage of colistin with less nephrotoxicity. However, the authors did not clearly mention optimal dosage of colistin. Therefore, the authors should mention association between the optimal dosage and efficacy or nephrotoxicity. Moreover, the authors should classify the optimal dosage based on patients’ background such as renal functions and types of infections. In general, colistin is administered with a loading dose for patients with normal renal functions. So, the authors need to describe the utility and advantage of the loading dose with a viewpoint of pharmacokinetics-pharmacodynamics. Finally, PRISMA was updated in 2021 (http://prisma-statement.org). So, the authors had better perform the review according to the updated PRISMA guidelines.

Author Response

Many thanks for providing positive feedback. We have corrected all comments by reviewer 2. Detail of each correction is given below.

Reviewer 1 Comment 1:

The authors systematically reviewed articles regarding the dosage of colistin, and stated the nephrotoxicity by colistin. This review is useful in the clinical settings due to a lack of data regarding the optimal dosage of colistin with less nephrotoxicity. However, the authors did not clearly mention optimal dosage of colistin. Therefore, the authors should mention association between the optimal dosage and efficacy or nephrotoxicity.

 Author’s Response to Reviewer 2 comment 1:

We thank the reviewer for their careful thoughts and constructive suggestion to improve our paper. Some more explanations have been added.

Reviewer 2 Comment 2:

Moreover, the authors should classify the optimal dosage based on patients’ background such as renal functions and types of infections. In general, colistin is administered with a loading dose for patients with normal renal functions. So, the authors need to describe the utility and advantage of the loading dose with a viewpoint of pharmacokinetics-pharmacodynamics.

Author’s Response to Reviewer 2 comment 2:

Thank you for your suggestion. Some more explanations have been added in the manuscript.

Reviewer 2 Comment 3:

Finally, PRISMA was updated in 2021 (http://prisma-statement.org). So, the authors had better perform the review according to the updated PRISMA guidelines.

Author’s Response to Reviewer 2 comment 3:

Thank you for the response. The methodology has been modified according to update PRISMA guidelines.

Round 2

Reviewer 2 Report

The comments by authors is appropriate against my questions.